# Obesity dysregulates the pulmonary antiviral immune response

Mark Almond[1,9], Hugo A. Farne [1,9], Millie M. Jackson[2,9], Akhilesh Jha[3], Orestis Katsoulis[2], Oliver Pitts[2], Tanushree Tunstall[1], Eteri Regis [1], Jake Dunning[4], Adam J. Byrne[1,5], Patrick Mallia[1], Onn Min Kon[1], Ken A. Saunders [6], Karen D. Simpson[6], Robert J. Snelgrove[1], Peter J. M. Openshaw [1], Michael R. Edwards[1], Wendy S. Barclay [7], Liam M. Heaney [8,9], Sebastian L. Johnston [1,9] & Aran Singanayagam [2,9] ✉

Obesity is a well-recognized risk factor for severe influenza infections but the mechanisms underlying susceptibility are poorly understood. Here, we identify that obese individuals have deficient pulmonary antiviral immune responses in bronchoalveolar lavage cells but not in bronchial epithelial cells or peripheral blood dendritic cells. We show that the obese human airway metabolome is perturbed with associated increases in the airway concentrations of the adipokine leptin which correlated negatively with the magnitude of ex vivo antiviral responses. Exogenous pulmonary leptin administration in mice directly impaired antiviral type I interferon responses in vivo and ex vivo in cultured airway macrophages. Obese individuals hospitalised with influenza showed dysregulated upper airway immune responses. These studies provide insight into mechanisms driving propensity to severe influenza infections in obesity and raise the potential for development of leptin manipulation or interferon administration as novel strategies for conferring protection from severe infections in obese higher risk individuals.

The overconsumption of calorie-rich processed foods and an increasingly sedentary lifestyle have led to a dramatic increase in obesity, now estimated to affect over 650 million globally[1]. An increased susceptibility of obese individuals to severe respiratory viral infections was initially highlighted during the 2009 H1N1 influenza pandemic where this comorbidity was consistently observed to be independently associated with adverse outcomes including requirement for hospitalisation and death[2–4]. Similar risks have been demonstrated in studies of seasonal influenza strains[5]. More recently, obesity has also emerged as a major risk factor for adverse outcome from coronavirus disease 2019 (COVID-19)[6].

Obesity may increase the risk of adverse outcomes from respiratory viral infections via a number of putative mechanisms including the influence of comorbid conditions (e.g. cardiovascular disease), direct consequences of altered lung mechanics (e.g. lung volume restriction)[7] or by having immunometabolic effects upon the pulmonary host response. Innate antiviral immune responses including the production of type I and III interferons (IFN) and subsequent induction of interferon-stimulated genes (ISGs) are a major mechanism of pulmonary mucosal protection against viruses such as influenza. Support for defective innate immunity as a mechanistic driver of obesity-related adverse consequences comes from animal studies where obese

[1]National Heart and Lung Institute, Imperial College London, London, UK. [2]Centre for Bacterial Resistance Biology. Section of Molecular Microbiology. Department of Infectious Disease, Imperial College London, London, UK. [3]Department of Medicine, University of Cambridge, Cambridge, UK. [4]Pandemic Sciences Institute, University of Oxford, Oxford, UK. [5]School of Medicine and Conway Institute of Biomolecular and Biomedical Research, University College Dublin, Dublin 4, Ireland. [6]Immunology Research Unit, GSK, Stevenage, UK. [7]Section of Virology, Department of Infectious Disease, Imperial College London, London, UK. [8]School of Sport, Exercise and Health Sciences, Loughborough University, Loughborough, UK. [9]These authors contributed equally: Mark Almond, Hugo A. Farne, Millie M. Jackson, Liam M. Heaney, Sebastian L. Johnston, Aran Singanayagam. ✉e-mail: a.singanayagam@imperial.ac.uk

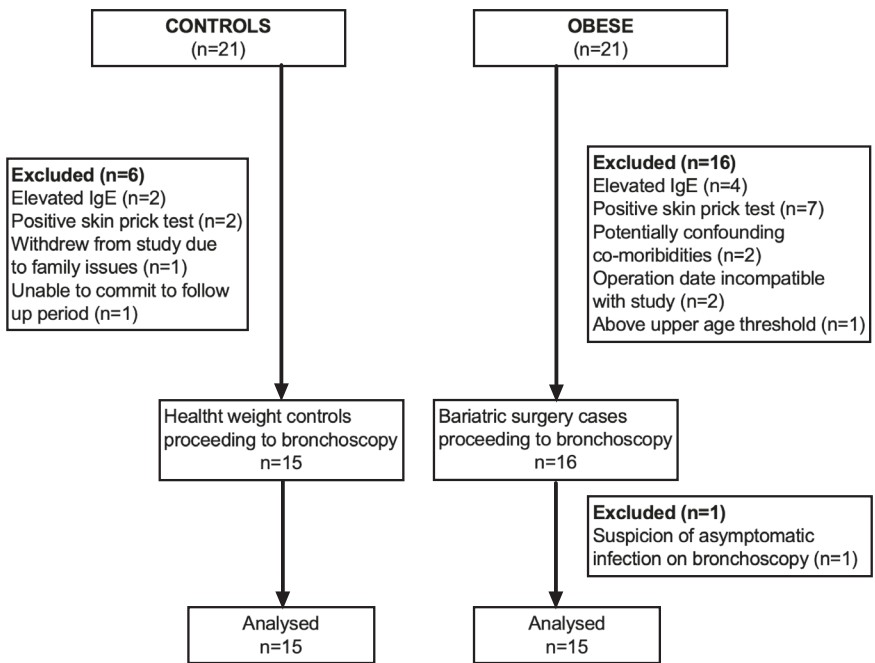

**Fig. 1 | Schematic diagram showing recruitment into the study.** Subjects were recruited as part of a case control study. Total subject numbers screened in each group and subjects excluded are shown.

mice show attenuated induction of type I IFN pathways associated with impaired virus clearance[8–10]. However, this has not been observed consistently across all influenza strains[10,11]. Whether a similar impairment in innate immune responses occurs in obese humans to drive adverse clinical outcomes from influenza is poorly understood. Moreover, the key type I and III IFN-producing cell types (e.g. macrophages, epithelial cells or dendritic cells) that may be affected and mechanisms that drive any potential defect are unknown.

Here, we utilise multi-compartment peripheral blood and airway sampling within a study of morbidly obese individuals undergoing bariatric surgery. This study design gave unique access to stable-state lower respiratory tract samples and allowed detailed investigation into inherent alterations of innate antiviral immunity in obese individuals relative to non-obese control subjects. We identify that obese individuals display deficient ex vivo type I and III IFN responses to influenza, a defect that occurs in airway macrophages but not other principal IFN-producing cell types (epithelial cells or dendritic cells). We elucidate that the airway metabolomic milieu is perturbed in obesity and that concentrations of prototypical adipokines, including leptin, are altered. Using functional experiments in mouse models of viral infection, we demonstrate that increased leptin is causally related to impairment of antiviral immunity in macrophages. Finally, we demonstrate that airway immune responses are dysregulated in obese individuals hospitalised with naturally acquired influenza infections. Taken together, these data implicate deficient airway innate immunity involving macrophages in the increased propensity to severe viral infections observed in obese individuals.

## Results

### Study subjects

The patient enrolment process for recruitment of subjects in this study is summarised in Fig. 1. There were 15 obese subjects and 15 non-obese control subjects included. There were no significant differences between the groups other than on anthropomorphic measures, as intended (Table 1). Of note, 2 of the 15 obese subjects had type 2 diabetes mellitus, both of which were treated with metformin only (not insulin dependent).

**Table 1 | Demographic and anthropometric characteristics in obese and non-obese subjects included in ex vivo experiments. Data represented as median (IQR) and analysed by two-tailed Mann–Witney *U* test or Fisher's Exact test**

| | Obese (*n* = 15) | Non obese (*n* = 15) | P value |
|---|---|---|---|
| Age | 46 (30–52) | 37 (30–50) | 0.62 |
| Gender (M:F) | 5:10 | 3:15 | 0.68 |
| Ethnicity | | | |
| White British | 11 | 8 | – |
| White other | 1 | 5 | – |
| Asian | 2 | 1 | – |
| Black | 1 | 1 | – |
| Height (m) | 1.67 (1.57–1.85) | 1.68 (1.59–1.73) | 0.91 |
| Weight (kg) | 144 (112–169) | 61 (56–73) | <0.001 |
| Body mass index (kg/m$^2$) | 49.1 (41.0–62.1) | 22.2 (20.6–24.6) | <0.001 |
| Skinfold thickness - biceps (mm) | 33 (21–38) | 8 (4–10) | <0.001 |
| Skinfold thickness - triceps (mm) | 42 (35–53) | 16 (12–20) | <0.001 |
| Skinfold thickness - suprailiac (mm) | 41 (37–57) | 17 (11–19) | <0.001 |
| Skinfold thickness - subscapular (mm) | 51 (36–61) | 11 (9–17) | <0.001 |
| Body circumference - neck (cm) | 42 (39–50) | 36 (34–38) | <0.001 |
| Body circumference - mid-upper arm (cm) | 41 (37–44) | 28 (25–28) | <0.001 |
| Body circumference - hip(cm) | 148 (131–160) | 99 (93–107) | <0.001 |
| Body circumference - waist (cm) | 126 (119–144) | 76 (70–81) | <0.001 |
| FEV$_1$ (L) | 3.07 (2.61–3.20) | 3.20 (2.91–3.67) | 0.17 |
| FVC (L) | 3.53 (3.24–3.96) | 4.04 (3.70–4.70) | 0.074 |
| FEV$_1$/FVC (%) | 101 (96–105) | 100 (96–105) | 0.67 |
| IgE (iU/mL) | 25 (7–66) | 15 (6–33) | 0.44 |
| Type 2 Diabetes mellitus | 2 | 0 | 0.14 |

## Obesity does not alter bronchial epithelial immune responses

Respiratory epithelial cells are the primary target cells for influenza infection[12] and impaired epithelial type I and III antiviral IFN responses have been reported in asthma and COPD, conditions similarly associated with susceptibility to severe viral infection[13–15]. We therefore initially hypothesised that obese individuals would display a similar impairment in epithelial antiviral immunity. In some recruited subjects, bronchial epithelial cell (BEC) cultures were unsuccessful; we therefore included 10 obese and 13 non-obese individuals in these experiments (Fig. 2a). Cells were stimulated ex vivo with the clinically relevant influenza viruses: A/Eng/195 (pandemic H1N1/09), A/Eng/691/10 (seasonal H3N2) and B/Florida (influenza B). No significant induction of type I IFN-α protein was observed for any of the three strains of influenza at 24 h (Fig. 2b). Type I IFN-β and -type III IFN-λ1 were significantly induced by H3N2 at 24 h with no induction observed for H1N1 or B/Florida viruses (Fig. 2c, d). The magnitude of induction of IFNs-β and -λ by H3N2 was no different between obese and non-obese individuals (Fig. 2c, d). Combined, these findings indicate that, unlike other disease states predisposed to severe influenza infection, obesity does not impair the epithelial antiviral response.

As airway inflammation may also drive severity of virus infections[16,17], we additionally evaluated the pro-inflammatory cytokine response in these experiments. At 24 h post-infection pH1N1/09 did not significantly induce IL-6, IL-8 or TNF, whereas H3N2 resulted in significant expression of all three cytokines in obese and non-obese subjects (Fig. 2e–g). B/Florida similarly induced all three cytokines in non-obese and obese subjects (Fig. 2e–g). As observed with the antiviral response, there was no difference in the induction of IL-6, IL-8 or TNF proteins between obese and non-obese individuals (Fig. 2e–g),

demonstrating that obesity also does not alter the epithelial pro-inflammatory response to influenza infection.

## Antiviral immune responses are impaired in bronchoalveolar lavage cells from obese individuals

Having observed that obese individuals did not display dysregulated epithelial immunity to influenza, we next focused on responses in BAL cells, comprising ~95% alveolar macrophages. Macrophages also have a key role in the innate antiviral immune response to infection[18] and similarly display deficient IFN responses in other conditions predisposed to severe viral infection such as asthma and COPD[19,20]. BAL cells obtained from obese and non-obese subjects were therefore infected ex vivo with H1N1/09, H3N2 and B/Florida (Fig. 3a). IFN-α was induced by all three viruses with significantly reduced induction observed in obese compared with non-obese individuals (Fig. 3b). IFN-β and IFN-λ were also induced by all three viruses in non-obese individuals but induction of IFN-β by H1N1 and IFN-λ by all three viruses failed to reach statistical significance and induction was significantly attenuated in obese compared with non-obese subjects (Fig. 3c, d). These data indicated that, in contrast to the epithelial response which is unaffected in obesity, BAL cells from obese individuals have an impaired potential to generate protective type I and III IFN antiviral responses. Despite this impairment of innate antiviral immunity, we did not detect any differences in BAL cell influenza M gene copies between obese and non-obese individuals at 24 or 48 h post-infection (Fig. 3e–g).

We additionally evaluated pro-inflammatory cytokine production in response to influenza infection in BAL cells. Baseline (medium-treated) BAL cells from obese individuals had significantly reduced spontaneous and post-infection production of IL-6, CXCL8/IL-8 and TNF compared to non-obese individuals. (Fig. 3h–j).

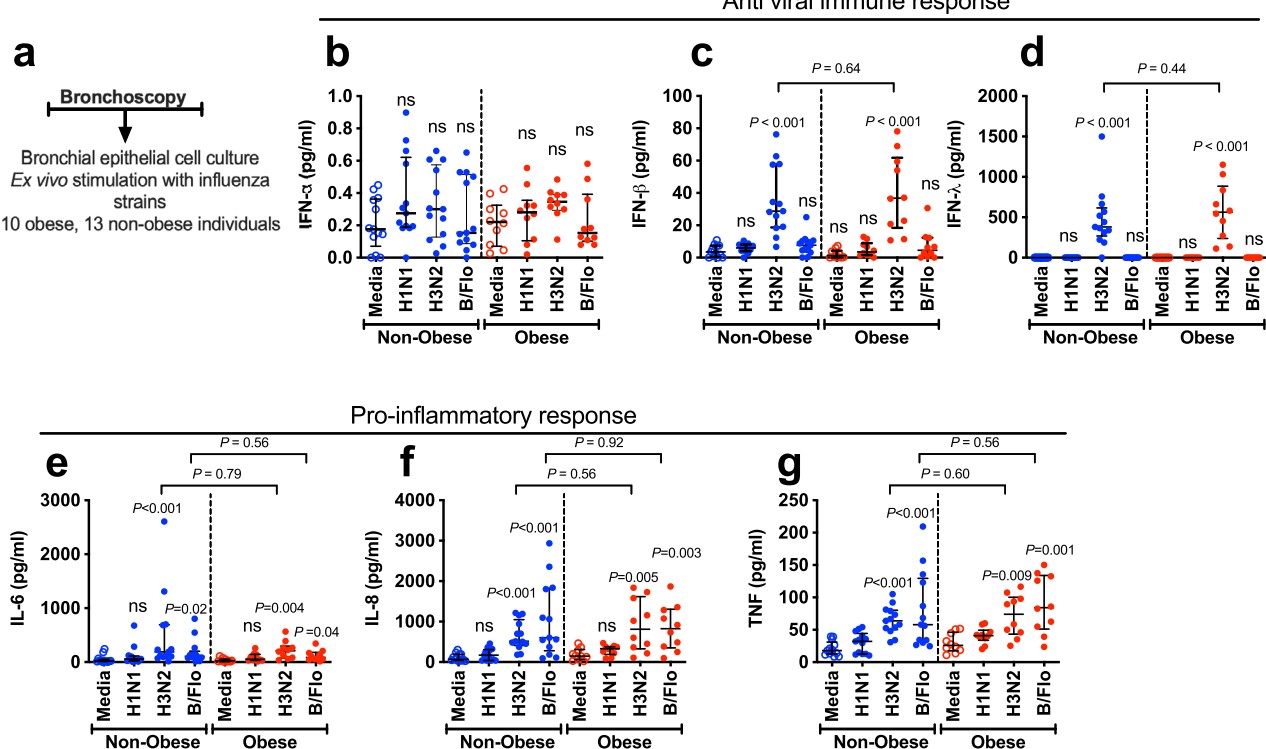

**Fig. 2 | Obesity does not alter bronchial epithelial cell antiviral or proinflammatory responses to influenza infection. a** Bronchial epithelial cells from 10 obese and 13 non-obese control subjects were cultured and infected ex vivo with H1N1/09 (H1N1), seasonal H3N2 and B/Florida (B/Flo) influenza viruses. Cell supernatants were collected at 24 h. **b** Interferon (IFN)-α, (**c**) IFN-β, (**d**) IFN-λ, (**e**) IL-6, (**f**) CXCL-8/IL-8 and (**g**) TNF protein concentrations were quantified by multiplex

ELISA. Graphs show datapoints for individual subjects with lines indicating median (IQR). Data analysed by Kruskal Wallis with Dunn's post-test or Mann–Whitney U test. Significance indicated above each group compares virus infected cells to medium-treated control cells. Brackets show comparisons between non-obese and obese subject groups. ns non-significant. Source data are provided as a Source Data file.

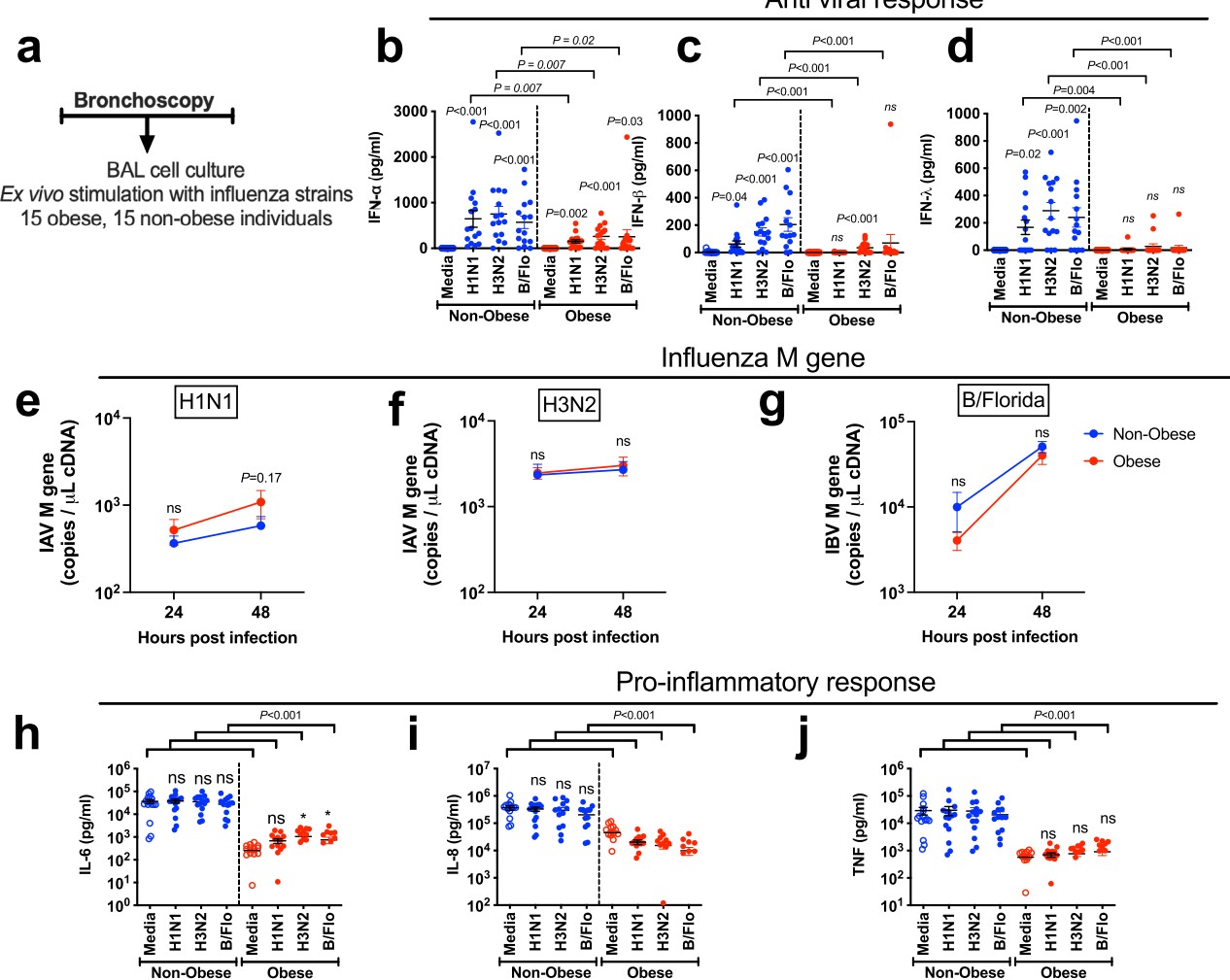

**Fig. 3 | Antiviral immune responses are impaired in bronchoalveolar lavage cells from obese subjects. a** Bronchoalveolar lavage (BAL) cells from 15 obese and 15 non-obese control subjects were cultured and infected ex vivo with H1N1/09 (H1N1), seasonal H3N2 and B/Florida (B/Flo) influenza viruses. Cell supernatants and lysates were collected at 24 and 48 h post-infection. **b** Interferon (IFN)-α, (**c**) IFN-β, and (**d**) IFN-λ. Influenza M gene mRNA expression was measured at 24 and 48 h following infection with (**e**), H1N1 (**f**) H3N2, and (**g**) B/Florida viruses. **h** IL-6, (**i**) CXCL-8/IL-8, and (**j**) TNF protein concentrations were quantified by multiplex ELISA. In **b**–**d** and **h**–**j** graphs show datapoints for individual subjects with lines indicating median (IQR). Significance indicated above each group compares virus infected cells to medium-treated control cells. In (**e**–**g**) data only measured in a subset due to sample dropout (RNA quality) and data expressed as median (IQR) and significance indicated above each time point compares non-obese with obese subjects. Data analysed by Kruskal–Wallis with Dunn's post test. Brackets show comparisons between non-obese and obese individual groups. ns non-significant. Source data are provided as a Source Data file.

## Obesity does not alter ex vivo antiviral responses in dendritic cells

Dendritic cells (DCs) are resident in the peripheral blood and lung and the plasmacytoid DC subset which express TLRs 7, 8 and 9 can be stimulated by influenza virus ssRNA to produce type I IFN[21]. Moreover plasmacytoid DCs (pDCs) are known to be particularly potent producers of type I IFN[22] We therefore studied whether obesity affected antiviral immune responses in these cells by isolating pDCs from peripheral blood by magnetic-activated cell sorting (MACS) (Supplementary Fig. 1a) followed by stimulation ex vivo with the same three influenza virus strains. There was no difference in DC purity between obese and non-obese subjects (Supplementary Fig. 1b). Significant induction of IFN-α, -β and -λ1 by H1N1 and B/Florida was observed in non-obese subjects and of IFN-α and -β by all three influenza strains in obese subjects with no differences in the magnitudes of induction in obese versus non-obese subjects (Supplementary Fig. 1c–e). Therefore, as observed for BECs, obesity does not impair the dendritic cell antiviral immune response.

Evaluation of pro-inflammatory cytokine responses in dendritic cells showed that IL-6 was induced by H1N1 and B/Florida in non-obese subjects and by all three viruses in obese subjects with no difference observed between the two subject groups (Supplementary Fig. 1f). For CXCL8/IL-8, induction occurred with H1N1 and B/Florida in obese and H1N1 in non-obese subjects with attenuation of responses for H1N1 observed in obese subjects (Supplementary Fig. 1g). TNF was induced by H1N1 and B/Florida in non-obese subjects and by all three viruses in obese subjects with no differences observed between the two subject groups (Supplementary Fig. 1h).

## Alterations of the airway metabolomic and adipokine milieu in obesity

Having observed a specific impairment of antiviral immunity within BAL macrophages from obese individuals, we next reasoned that the metabolomic milieu within the obese airways would be altered and may contribute to the observed ex vivo immune dysregulation. We therefore profiled BAL fluid metabolite abundances using Ultrahigh

Performance Liquid Chromatography-Tandem Mass Spectrometry (UPLC-MS/MS) to gain insight into potential mechanisms that may be driving impaired immunity in obesity (Fig. 4a). Unsupervised Principal Components Analysis demonstrated a partial separation of obese and non-obese groupings (Supplementary Fig 2). This was followed by a supervised analysis where Orthogonal Projections to Latent Structures Discriminant Analysis (OPLS-DA) revealed differences between obese and non-obese individuals (Fig. 4b). Differential abundance analysis performed via visualisation of the volcano plot identified 17 metabolites with significantly altered responses with the majority ($n = 15$) downregulated and two metabolites (adenosine monophosphate (AMP) and glycerol) upregulated in BAL from obese individuals (Fig. 4c). Heat map of the 50 metabolites with the highest peak area responses detected in BAL revealed a similar pattern, with all but three being elevated in non-obese participants, with only AMP, glycerol and mannitol/sorbitol being elevated in obese participants (Fig. 4d and Supplementary Fig. 4). Enrichment analysis indicated a range of different metabolites that were enriched within the cohort, including seleno-amino acid, methionine and fatty acid metabolic pathways in obese compared with non-obese participants (Fig. 4e).

Fatty acid metabolism has been shown to influence antiviral responses as dysregulated cholesterol biosynthesis can impact type I IFN signalling[23]. Obesity is linked to fatty acid metabolism through adipokines such as leptin which can stimulate fatty acid oxidation via AMP-activated protein kinase[24]. Since AMP was one of the two metabolites significantly upregulated in BAL fluid from obese participants, we therefore further focused on a targeted evaluation of adipokine concentrations. These cell signalling molecules are produced by adipose tissue and can exert pleiotropic effects upon immune function. Leptin, specifically, has been shown to impair macrophage-mediated immune responses to bacterial pathogens[25] and we therefore postulated that similar effects could occur in the context of antiviral immunity. We found that leptin concentrations were augmented in obese versus non-obese individuals in both the upper (nasosorption) and lower (bronchosorption and BAL) airways (Fig. 4f). Measurement of other adipokines showed reduced adiponectin concentrations in BAL and nasosorption but not bronchosorption (Supplementary Fig. 3a) and reduced concentrations of visfatin in BAL but not nasosorption or bronchosorption (Supplementary Fig. 3b) in obese compared with non-obese individuals. There were no significant differences in resistin concentrations in any sample type (Supplementary Fig. 3c).

Furthermore, bronchosorption concentrations of leptin negatively correlated with the magnitude of BAL cell IFN-β responses to all three influenza strains tested in our ex vivo experiments, with greater concentrations of leptin being significantly associated with weaker induction of IFN-β by each virus strain (Fig. 4g). Bronchosorption concentrations of leptin also significantly positively correlated with BAL AMP levels (Fig. 4h). This indicated a possible causal link between raised leptin concentrations and impaired antiviral immunity in obesity, potentially through perturbed fatty acid metabolism. By contrast, bronchosorption concentrations of adiponectin, visfatin and resistin showed no significant correlations with ex vivo BAL cell IFN-β responses to all three influenza strains tested (Supplementary Fig. 3d–f).

### Exogenous leptin administration reduces airway type I IFN responses to influenza infection in mice
Next, to elucidate if suppression of IFN responses in obesity is functionally related to increased airway leptin concentrations, we used pulmonary recombinant leptin protein administration (at a previously reported dose[26]) in mice (Fig. 5a) to mimic the increased airway concentrations observed in human subjects and permit study of direct causal effects upon antiviral immunity. Consistent with the known pro-inflammatory effect of leptin[27], administration of exogenous leptin

alone increased lung mRNA expression of *IfnB* and the antiviral ISGs *2′-5′Oas*, *Viperin* and *Pkr* compared to vehicle-treated controls (Fig. 5b). All four of these genes were also induced in lung tissue by influenza X31 (a mouse adapted H3N2 strain) at 6 h post-infection. Leptin administration prior to influenza infection attenuated virus-induced expression of all four of these antiviral immune genes (Fig. 5b), indicating that leptin is functionally related to attenuated pulmonary type I IFN immune responses. This was associated with a later augmentation in pro-inflammatory responses at 72 h post-infection with increased BAL neutrophils, activated neutrophil subsets (CD63+ and CD64+) and pro-inflammatory cytokines IL-1β, IL-6 and TNF observed in mice treated with leptin prior to infection compared to vehicle-treated infected control mice (Fig. 5c–f).

Previous studies have reported that raised suppressor of cytokine signalling (SOCS)-3 in peripheral blood mononuclear cells from obese individuals is associated with reduced production of type I IFNs in response to TLR-agonist stimulation[28]. Accordingly, we found that leptin administration in mice increased lung expression of *Socs3* mRNA (Fig. 5g). In conjunction with the concomitant reduced induction of *IfnB* and ISGs upon viral infection this suggests that high leptin levels in obesity may promote increased expression of SOCS-3, a known negative regulator of JAK-STAT-mediated type I IFN signalling, to attenuate the early antiviral immune response during influenza infection.

### Leptin treatment impairs ex vivo type I IFN responses in mouse macrophages
We next determined whether the suppressive effect of leptin upon pulmonary antiviral immunity was mediated through effects on airway macrophages, given our prior data of a specific defect in this cell type from obese human subjects (Fig. 3). We therefore isolated BAL macrophages from mice and infected them ex vivo with influenza, in the presence or absence of exogenous leptin treatment at 320 μg/ml (mimicking the dose used in vivo) and ten-fold lower 32 μg/ml concentrations (Fig. 5h). Influenza infection induced *IfnB* and the ISGs *2′-5′ Oas* and *Pkr* in macrophages (Fig. 5i). Leptin treatment (320 μg/ml) significantly impaired virus-induction of *IfnB and Pkr* with a trend towards reduced *2′-5′ Oas* (Fig. 5i). As observed in vivo with whole lung expression, leptin treatment also increased *Socs3* mRNA specifically in macrophages (Fig. 5j). These data corroborated our human ex vivo findings and confirmed a specific defect of macrophage immunity associated with augmented airway leptin.

### Dysregulated airway immune responses in obese individuals hospitalised with influenza
To confirm the clinical relevance of our ex vivo human and ex vivo and in vivo animal findings, we studied upper airway immune responses in the Mechanisms of Severe Acute Influenza Consortium (MOSAIC) study which recruited adult patients hospitalised with clinical influenza presenting to hospitals in London and Liverpool (UK) during the winters of 2009/10 and 2010/11 (periods of intense influenza activity in the UK). The MOSAIC study was associated with the larger multicentre FLU-CIN (Influenza Clinical Information Network) cohort study in which obesity has already been clearly shown to be independently associated with adverse outcomes[29] Of 133 patients recruited to the study, $n = 27$ (20.3%) were classified as obese (BMI > 30; Fig. 6a). Baseline demographic and clinical data of obese and non-obese individuals are shown in Table 2.

There were no differences in nasopharyngeal virus loads between obese and non-obese patients during the acute (24 and 48 h after hospitalisation) or recovery (>4 weeks) phases of influenza infection (Fig. 6b). There was also no difference in nasopharyngeal IFN-α or IFN-γ concentrations between the two groups but obese patients had increased concentrations of IFN-β at 24 and 48 h after presentation, compared to non-obese patients (Fig. 6c). Pro-inflammatory cytokine

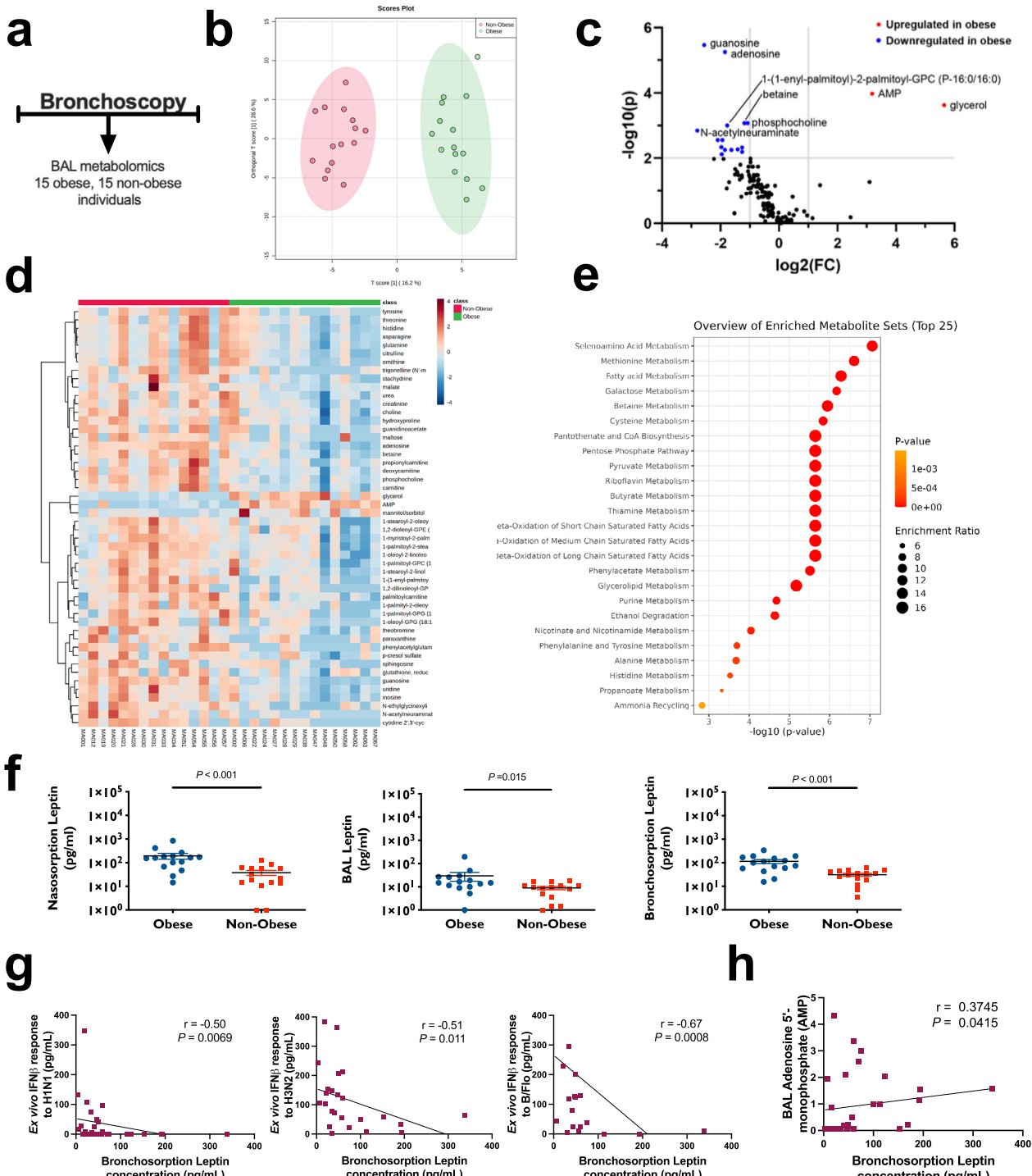

**Fig. 4 | Altered airway metabolomic milieu in obesity. a** Bronchoalveolar lavage (BAL) fluid from 15 obese and 15 non-obese control subjects was collected and processed for metabolomics analysis. **b** Orthogonal projections to latent structures-discriminant analysis (OPLS-DA) showing separation of obese (green) and non-obese (red) subjects. **c** Differential abundance analysis, performed via visualisation of the volcano plot. Data were generated using an unpaired *t*-test (2-sided) and *p* values refer to the unadjusted and -log₁₀ transformed value. **d** A heat map clustered for obese (green) and non-obese (red) subjects to show the top 50 metabolite correlations. An enlarged version of this plot to enable the reading of metabolite annotations can be found in Supplementary Fig. 4 (**e**) Enrichment analysis identifying the top 25 representative metabolic pathways significantly enriched in obese vs. non-obese BAL fluid. Data were computed using the R package *globaltest* which applies a generalised linear model to estimate a *Q-statistic*, with *p* values (2-sided) referring to the unadjusted and -log₁₀ transformed value. **f** Measurements of leptin in obese vs non-obese subjects in upper (nasosorption, left) and lower (bronchosorption, right, BAL fluid centre) airway samples shown as individual values and median (solid horizontal line) and IQR. **g**, **h** Correlations of bronchosorption leptin concentrations with (**g**) the magnitude of ex vivo BAL cell IFN-β responses to H1N1 (left), H3N2 (centre), and B/Flo (right) influenza strains and (**h**) BAL fluid AMP concentrations. Data in F analysed by Mann–Whitney *U* test. Data in G-H analysed by Spearman's rank correlation test (two sided). Source data are provided as a Source Data file.

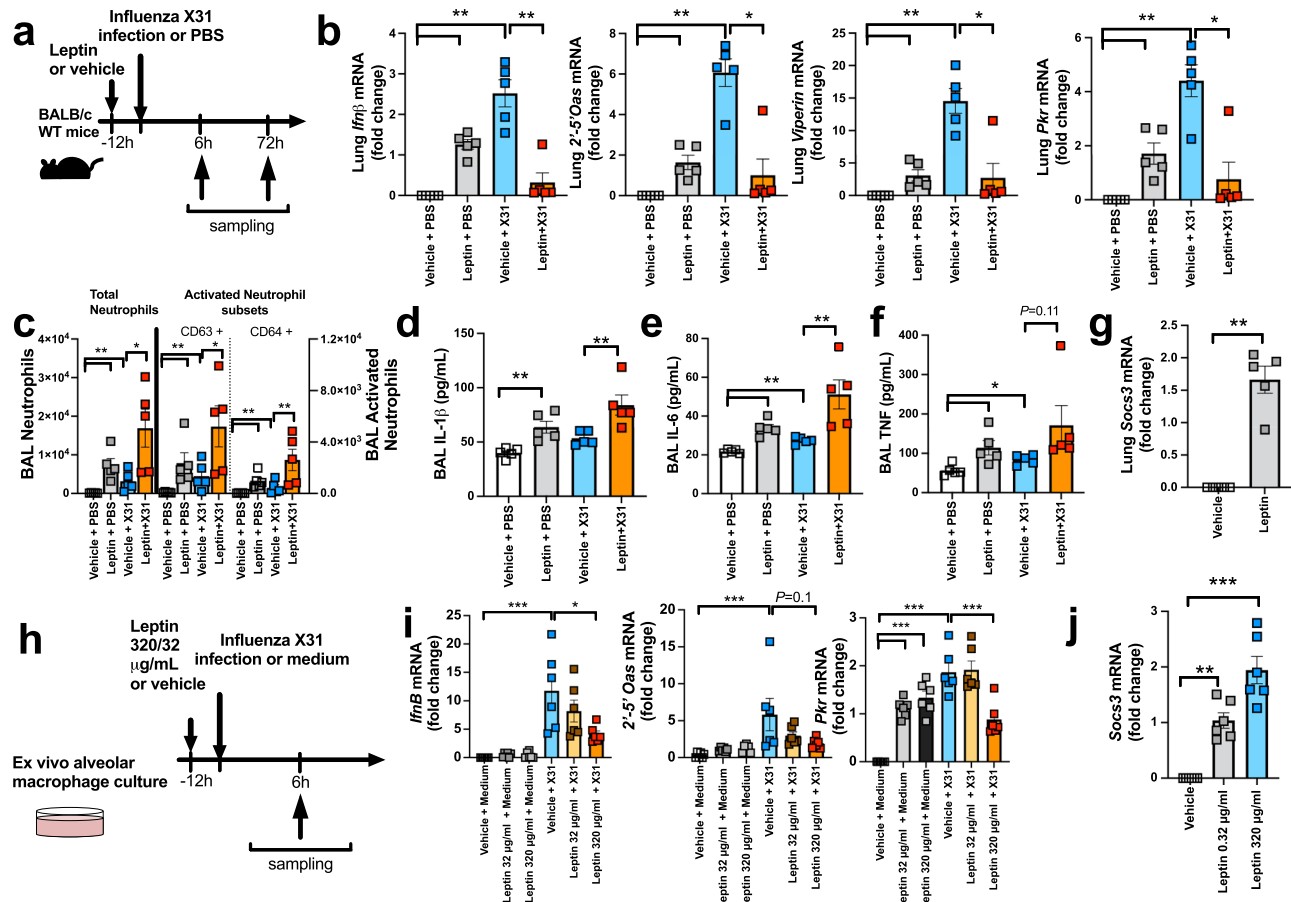

**Fig. 5 | Pulmonary leptin administration reduces airway immune responses to influenza infection in mice. a** BALB/c mice were treated with intranasal leptin protein (16μg) or vehicle control, 12 h prior to infection with Influenza X31 or PBS control. **b** *IfnB, 2'-5'Oas, Viperin* and *PKR* lung mRNA expression measured by qPCR at 6 h post-infection. **c** BAL neutrophils and activated neutrophil subsets (CD63+ and CD64+) enumerated by flow cytometry at 72 h post-infection. BAL concentrations of (**d**) IL-1β, (**e**) IL-6 and (**f**) TNF measured by ELISA at 72 h post-infection. **g** Lung *Socs3* mRNA expression at 18 h following leptin administration. **h** BAL macrophages were harvested from naïve untreated mice and cultured ex vivo followed by treatment with leptin protein (32 and 320 μg/ml) for 12 h before infection with Influenza X31. **i** *IfnβB, 2'–5'Oas, and PKR* mRNA expression in cell lysates at 6 h post-infection. **j** *Socs3* mRNA expression at 18 h following leptin administration. Data are presented as mean (±SEM) for *n* = 5 mice per group in **b**–**g** and *n* = 5 for **i, j**. Statistical significance analysed using one-way ANOVA with Bonferroni post-test. *\*P* < 0.05, *\*\*P* < 0.01, *\*\*\*P* < 0.001. Source data are provided as a Source Data file.

concentrations were also increased in nasopharyngeal aspirates from obese patients including IL-1, IL-6, CXCL8/IL-8 at 24 and 48 h, CXCL10/IP-10 at 24 h and TNF at 48 h after presentation, all compared to non-obese patients (Fig. 6c). By contrast, there were no significant differences in serum concentrations of any of these cytokines between obese and non-obese patients (Fig. 6d). Collectively, these findings indicate that, during acute hospitalised influenza infection, immune dysregulation occurs locally within the upper airway mucosa but not centrally within the systemic circulation.

## Discussion

The mechanisms that underlie susceptibility to severe respiratory viral infections in individuals with comorbidities such as obesity remains a major area of interest in the field. In this study we combined human and animal studies to demonstrate that obesity is associated with a specific impairment of antiviral type I and III IFN responses to influenza infection within bronchoalveolar lavage (BAL) macrophages. Our experiments indicate a mechanism for impairment of antiviral immunity in the obese lung occurring through increased airway leptin concentrations which we observed were increased in the airways of obese people and which directly inhibited protective pulmonary antiviral immune responses when administered to mice.

In recent years, there has been intense focus on susceptibility to SARS-CoV-2 infection with type I and III interferon immunopathology being a major pathway of interest[30]. However, influenza remains a significant cause of morbidity, mortality and healthcare costs worldwide with numbers of positive cases rising exponentially recently due to increased social interactions following easing of COVID-19 lockdown measures. Therefore, better understanding of mechanisms driving susceptibility to influenza in at-risk individuals remains a major global research priority. Obesity has previously been identified to be a strong risk factor for severe influenza infections during previous pandemic and seasonal outbreaks[2-5] but the reasons underlying this heightened susceptibility have been unclear. Although vaccination is recommended for high-risk individuals, including those with BMI > 40, these approaches do not confer complete protection and new approaches using antiviral or immunomodulatory therapies are urgently needed.

Production of type I (-α and -β) and type III (-λ) IFNs are a major host protective mechanism to counteract respiratory virus infections and, in their absence, mice succumb to influenza infection[31]. Both diet- and genetically-induced obese mice (leptin- or leptin receptor-deficient) suffer greater lung damage and higher mortality from influenza than non-obese controls[31]. Detailed study of these mouse models has

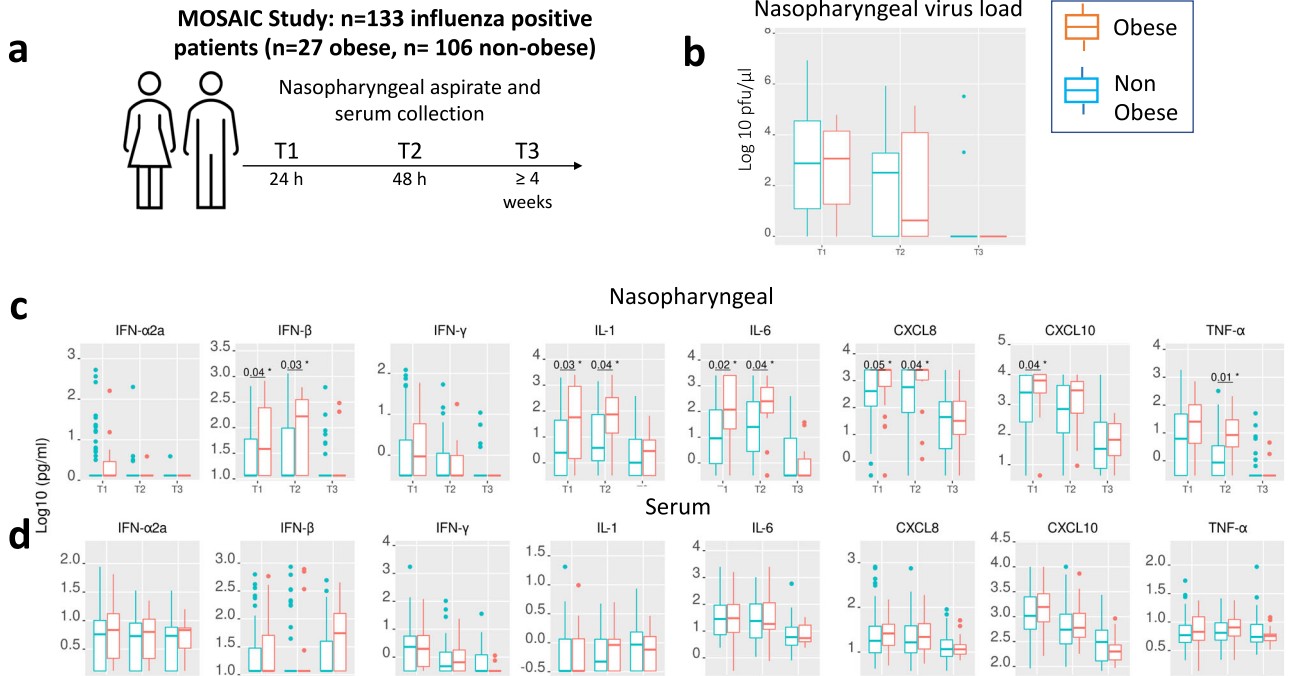

**Fig. 6 | Altered upper airway immune responses in obese adult patients hospitalised with influenza infection. a** Schematic of the Mechanisms of Severe Acute Influenza Consortium (MOSAIC) study highlighting timepoints from hospitalisation of study samples. **b** Nasopharyngeal virus load between obese and non-obese patients. **c** Nasopharyngeal aspirate and (**d**) serum multiplex immune mediators comparing obese and non-obese patients. Data from 133 influenza positive adults (27 obese, 106 non-obese) compared using Mann–Whitney test (two-tailed). Box and whisker plots show median (line within box), interquartile range (box) and 1.5 x IQR (*whiskers*) *P < 0.05. Source data are provided as a Source Data file.

repeatedly found attenuated type I IFN production in obese mice infected with influenza[9,10,32–37]. Despite this evidence from animal studies, there are very few studies that directly investigate obesity-related antiviral responses in human subjects. Limited case series report greater viral replication with prolonged shedding in obese subjects[38,39]. Studies also indicate that PBMCs from obese individuals show impaired type I IFN responses to TLR3 agonist stimulation[40,41]. However, examination of peripheral cell responses is an imprecise surrogate for responses within the lower respiratory tract (the primary site of influenza pathology). Obtaining lower airway samples from stable-state morbidly obese individuals to investigate pulmonary-specific immune responses is technically challenging since bronchoscopy under conscious sedation may be considered risky in these individuals. Our study therefore utilised a unique design whereby lower airway samples were taken directly from obese subjects during general anaesthesia being administered as part of a bariatric surgery procedure. This opportunistic approach allowed us to interrogate the ex vivo antiviral immune response within the two major lung resident IFN-producing cell populations, epithelial cells and macrophages. We identified that type I and III IFN responses are specifically impaired in BAL cells (comprised predominantly (~95%) of macrophages) from obese individuals but unaffected in bronchial epithelial cells (major producers of IFN-β and IFN-λ) or plasmacytoid dendritic cells (major producers of IFN-α). We and others have reported similar defects in BAL cells from other high-risk individuals including those with asthma and COPD[20,42] and recent single-cell transcriptomic data from intubated COVID-19 patients indicates attenuated type I IFN in obese versus non-obese subjects[43]. Alveolar macrophages are well known to play a key role in mediating immune responses to influenza and their ablation in mouse models worsens influenza-mediated pathology and mortality[44]. It has also been reported that alveolar macrophages are an important source of type I IFNs during influenza infection, as ex vivo IFN-α and -β induction was reduced in alveolar macrophages but not

pDCs sorted from the lungs of diet-induced obese mice[9]. Our data supports the central role played by these cells in innate immunity and highlight that their antiviral responses to influenza infection are attenuated in obese individuals.

Macrophage immune function can be reprogrammed through metabolomic alterations and we therefore hypothesised that the obese pulmonary microenvironment may be impacting upon type I IFN pathways to dampen innate antiviral immunity. Analysis of the airway metabolome in our subjects revealed profound alterations associated with obesity including dysregulation of fatty acid metabolism, a pathway that has been mechanistically linked to antiviral immunity[23,45]. AMP was one of two metabolites significantly upregulated in BAL from obese subjects and the adipokine leptin, through activation of AMP-activated kinase, is known to directly stimulate fatty-acid oxidation[24]. Accordingly, we identified increased concentrations of leptin in the obese airway which positively correlated with AMP concentrations and negatively correlated with the magnitude of ex vivo BAL cell type I IFN response, suggesting that leptin may directly impact upon antiviral pathways through perturbation of fatty acid metabolism. To investigate a functional relationship, we studied the effect of pulmonary recombinant leptin protein administration in mice to mimic the increased concentrations observed in the obese human airway and our results indicate a direct causal role for leptin in attenuating antiviral immune responses. Leptin is a non-glycosylated hormone of 146 amino acids that is synthesised by adipose cells in response to food intake and plays a central role in appetite and body weight homeostasis. It is increasingly recognised to have a regulatory role in metabolism-immune system interplay and exerts its effects upon immune cells through the cell surface leptin receptor (LepR)[46]. Leptin can increase oxidative stress in macrophages and in a diet-induced obesity influenza mouse model, hyperleptinaemia was associated with increased viral spread, inflammation and mortality, consequences that could be reversed by anti-leptin antibody administration[47] Surprisingly a greater number of

**Table 2 | Demographic and clinic characteristics in obese and non-obese subjects within the MOSAIC study**

|  | Obese (%) | Non obese (%) | P value |
|---|---|---|---|
| Total numbers | 27 (20.3) | 106 (79.7) | – |
| Demographics |  |  |  |
| Female | 17 (63.0) | 47 (44.3) | 0.090 |
| Age years |  |  |  |
| 18–30 | 5 (18.5) | 32 (30.2) | 0.34 |
| 31–45 | 13 (48.1) | 33 (31.1) | 0.12 |
| 46–60 | 9 (33.3) | 25 (23.6) | 0.33 |
| >60 | 0 (0) | 16 (15.1) | 0.041 |
| Ethnicity |  |  |  |
| White | 19 (70.4) | 70 (66.0) | 0.82 |
| Asian/Asian British | 2 (7.4) | 7 (6.6) | 1.0 |
| Black/Black British | 3 (11.1) | 19 (17.9) | 0.56 |
| Chinese | 3 (11.1) | 10 (9.4) | 0.73 |
| Co-morbidities |  |  |  |
| Asthma | 11 (40.7) | 32 (30.2) | 0.36 |
| Diabetes | 4 (14.8) | 10 (9.4) | 0.486 |
| Cardiovascular disease | 9 (33.3) | 20 (18.9) | 0.126 |
| Current smoker | 7/23 (30.4) | 35/85 (41.2) | 0.476 |
| Ex-smoker | 7/23 (30.4) | 14/85 (16.5) | 0.15 |
| Seasonal vaccine | 4/9 (44.4) | 28/46 (60.1) | 0.47 |
| H1N1 vaccine | 2/8 (25) | 13/24 (54.2) | 0.23 |

Data represented as n(%) and analysed by two-tailed Fisher's exact test.

metabolites were upregulated in BAL from non-obese subjects. The reasons for this observation are unclear and further independent validation in larger studies is needed to corroborate our findings.

Mechanistically, we identified that airway leptin augmentation induces expression of SOCS3 in macrophages, a known negative regulator of type I IFN pathways via inhibition of JAK/STAT signalling. Our data are supported by previous studies linking elevated leptin with induction of systemic SOCS3 expression in obese individuals, as a mechanism of negative feedback[46,48]. Genetic silencing of SOCS3 but not SOCS1 in PBMCs from obese individuals can boost type I IFN immunity and exogenous leptin administration also induces SOCS3 and inhibits type I IFN responses to TLR3 agonist stimulation in PBMCs[28]. cAMP has also been shown to upregulate SOCS3 in endothelial cells[49]. Our data therefore implicate the leptin-AMP-SOC3 axis as an important determinant of antiviral immunity in the airway that may drive susceptibility to severe influenza infections in morbidly obese individuals. Whether this pathway is similarly dysregulated in overweight or non-morbidly obese individuals remains unknown. Given the strong correlation between BMI and serum leptin concentrations reported in multiple studies, we would anticipate that airway leptin concentrations would be similarly correlated with BMI and therefore we postulate that a continuum (i.e. the same effect would be present but to a lesser degree) would be observed in terms of antiviral immune suppression. Further studies are needed to determine this.

Finally, to study the clinical relevance of our findings, we carried out immune analyses within samples from a 'real-world' cohort of adults hospitalised with influenza. Upper airway in vivo IFN responses were not deficient in hospitalised obese patients, which corroborated the results of our in vitro epithelial cell influenza infection experiments. Furthermore, the lack of differences in serum cytokine levels between obese and non-obese patients hospitalised with influenza were in keeping with results from ex vivo simulation of circulating pDCs. The MOSAIC study did not routinely have access to BAL data and therefore our results which utilised ex vivo stimulation of BAL cells demonstrating a type I and III IFN deficiency in obese individuals helps to provide a mechanistic understanding of why obese subjects are more susceptible to viral infections. We observed increased upper respiratory proinflammatory responses associated with obesity in this cohort, an effect that was also mimicked at late timepoints in our mouse model of leptin administration with influenza infection. There is likely to be a delay between virus infection and hospital presentation which typically occurs at the peak of symptom severity (around day 5–6 after viral inoculation in human influenza challenge models[50]). Therefore, the initial impairment of antiviral responses observed in macrophages ex vivo could drive an early presymptomatic increase in viral replication which may subsequently (by day 5–6) drive later accentuation of antiviral and proinflammatory airway responses. We have observed a similar pattern of antiviral immune expression in asthma where type I IFN deficiency is observed in BECs and BAL macrophages ex vivo[13,19,42] but in vivo type I IFN and pro-inflammatory cytokine expression is increased at day 4 after experimental RV challenge[51]. It should also be noted that COVID-19 studies have demonstrated disparate interferon expression in the upper and lower respiratory tract during acute disease[52].

In conclusion, our study uncovers insight into mechanisms driving susceptibility to severe influenza infections in obese individuals. Future work should focus on whether sustained weight loss leads to a restitution of this impaired antiviral immunity, especially given that epidemiological evidence indicates that the clinical risk of influenza infection diminishes following bariatric surgery[53] and impaired mononuclear cell type II IFN responses in obese individuals can be corrected by weight loss[54]. As a preventable condition, treatment of obesity itself must remain a priority and is potentially the most effective method of reducing risk of severe viral infections. However, this is not achievable for many individuals and the findings of our study open up the potential for leptin manipulation or IFN administration as novel strategies for conferring protection from severe infections.

## Methods
### Ethics statement
Human bronchoscopy study: the study received ethical approval from the Surrey Borders Research Ethics Committee (approval number 12/LO/1812). Informed consent was obtained from all participants.

The Mechanisms of Severe Acute Influenza Consortium (MOSAIC) study of hospitalised influenza infections: the study was approved by the NHS National Research Ethics Service, Outer West London REC (09/H0709/52, 09/MRE00/67). Informed consent was obtained from all participants.

Animal experiments: all animal work was performed under the authority of the UK Home Office outlined in the Animals (Scientific Procedures) Act 1986 after ethical review by Imperial College London Animal Welfare and Ethical Review Body (project licence PP4051423).

### Study design and participants
Morbidly obese individuals (body mass index (BMI) $\geq$ 35 kg/m$^2$) were prospectively recruited from the bariatric surgery service at Imperial College Healthcare NHS Trust as part of a case-control study. Healthy non-obese control subjects (BMI 20–25 kg/m$^2$) were matched for age, gender and ethnicity. Exclusion criteria included age over 60, atopy defined as skin prick test positivity to one or more of nine common aeroallergens, obstructive airways disease (asthma or COPD), history of smoking within the last 6 months or smoking history > 20 pack years, inhaled medication, statin or thiazolidinedione treatment (any of which could confound interferon (IFN) responses), respiratory tract infection within the preceding 3 months and pregnancy.

### Sample processing and cell culture
All subjects underwent anthropometric characterisation and clinical sampling including blood, nasal synthetic absorptive matrix (SAM) sampling (nasosorption, to sample airway lining fluid), and

bronchoscopy with bronchosorption, bronchial brushings, and BAL, as previously described[55,56].

**BAL macrophages:** BAL fluid was filtered, centrifuged at 200 RCF for 10 min. The supernatant was removed and stored at −80ºC for metabolomic/adipokine measurement. The remaining cell pellet (comprising>95% macrophages) was resuspended in medium. Cells were then washed ain PBS, resuspended at a final concentration of $2 \times 10^6$ cells/mL and then plated in a 96-well plate at $2 \times 10^5$ cells per well.

Bronchial epithelial cells: brushed epithelial cells were dislodged into bronchial epithelial cell growth medium (BEGM, Lonza). Cells were placed into a T25 culture flask and incubated at 37ºC with 5% CO2 until 95% confluency. Cells were passaged twice before being seeded onto 24 well plates at $5 \times 10^5$ cells per well.

Plasmacytoid dendritic cells: plasmacytoid and conventional dendritic cells (pDCs and cDCs respectively) were isolated from PBMCs using a commercial human DC isolation kit according to manufacturer's instructions (Miltenyi Biotec, Germany). The purity of the positively selected cell fraction containing the enriched DCs was confirmed by flow cytometry. Cells were plated in 96 well plates at 20,000 cells per well.

## Ex vivo virus infection experiments

BAL macrophages, BECs or DCs were infected with influenza viruses, A/Eng/195, A/Eng/691/10 or B/Florida, at a multiplicity of infection (MOI) of 3 or medium control. After one hour the viral inoculum was removed and fresh medium was added. Cell supernatants and lysates were subsequently harvested at 24 and 48 h. Supernatants and lysates were stored at −80 °C until protein quantification or RNA extraction respectively.

## Influenza infection and treatment of mice

In vivo *protocols:* Female mice (6–8 weeks of age) on a BALB/c background purchased from Charles River UK Laboratories were used for all animal studies. Mice were housed in individually ventilated cages under specific pathogen-free conditions.

Mice were treated intranasally under isofluorane anaesthesia with 16 ng of recombinant mouse leptin (Sigma Aldrich) in 50 µl of phosphate-buffered saline (PBS) or vehicle (PBS) control, as previously described[26]. Twelve hours after leptin treatment, mice were intranasally infected with influenza virus strain X31 ($1 \times 10^5$ pfu in 50 µl of PBS) or PBS control. Mice were euthanized with intraperitoneal overdose administration of pentobarbitone solution.

## In vitro mouse alveolar macrophage experiments

Naive mice underwent BAL with PBS supplemented with 2.5 mM EDTA. BAL cells were resuspended in 10% RPMI medium, seeded at a density of $5 \times 10^4$ cells per well and then incubated at 37 °C with 5% $CO_2$ for 2 h to allow adherence of macrophages. Cells were treated with leptin at 32 µg/ml and 320 µg/ml concentrations or medium control for 12 h before infection with influenxa X31 (MOI 1.0). Cell lysates were harvested at 6 h post-infection for downstream mRNA expression analysis.

## The Mechanisms of Severe Acute Influenza Consortium study of hospitalised influenza infections

Real-world data from obese and non-obese individuals with PCR-confirmed influenza were drawn from the multi-site, prospective, observational Mechanisms of Severe Acute Influenza Consortium (MOSAIC) study[57,58]. Detailed clinical and demographic data were recorded for all participants. BMI was calculated where height and weight data were available, defining obesity as BMI ≥ 30. Patients were also classified as obese if BMI could not be calculated but 'obese' was documented in the case report forms used to collect demographic data. Nasopharyngeal aspirates and swabs were taken within 72 h of

admission and used to confirm influenza infection by PCR. Out of a total of 133 patients, 120 were influenza A positive (116 H1N1, 3 H3N2 and 1 undetermined), 12 were influenza B positive and 1 had influenza A + B coinfection. Nasopharyngeal aspirates and blood samples were taken within 24 h of admission (as previously described[57,58]). Time 1 (T1) samples were collected within 24 h of admission, T2 samples within 48 h of admission and T3 samples in convalescence ≥4 weeks after presentation.

## Protein assays

IFN-α2a, IFN-β, IFN-λ1, IL-6, IL-8, and tumour necrosis factor (TNF) proteins were quantified in cell supernatants from ex vivo experiments using Meso Scale Discovery (MSD) assays (Mesoscale Discovery, USA). Adipokines were quantified in nasal lining fluid (nasosorption), bronchial lining fluid (bronchosorption) and BAL samples by Luminex immunoassays (Luminex, USA). Cytokine protein concentrations in mouse BAL were assayed using commercial 'duoset' enzyme-linked immunosorbent assay kits (Biotechne, UK). MSD assays were used to measure all immune proteins in nasal and serum samples from the MOSAIC study cohort.

## RNA and quantitative PCR for measurement of immune gene expression

Total RNA was extracted from the right upper lobe of mouse lung and placed in RNA later, prior to RNA extraction and cDNA synthesis using the Omniscript RT kit (Qiagen, UK). Quantitative PCR was carried out using previously described specific primers and probes for each gene of interest[59] and normalised to 18 S rRNA housekeeping gene. $2^{-\Delta\Delta Ct}$ (fold-change) was used to calculate gene expression relative to the control group. Primer/Probe sequences for mouse genes were as follows.

*Ifnb Forward:* CCATCATGAACAGGTGGAT; *Ifnb* Reverse: GAGAGGGCTGTGGTGGAGAA, *Ifnb* Probe: CTCCACGCTGCGTTCCT GCTGTG.

*2'-5' Oas* Forward: TCCTGGGTCATGTTAATACTTCCA, *2'-5' Oas* Reverse: CCCCAGGGAGGTACATTCCT, *2'-5' Oas* Probe:CAAGCCTG ATCCCAGAATCTATGCCATC;

*Viperin* Forward. CGAAGACATGAATGAACACATCAA. *Viperin* Reverse: AATTAGGAGGCACTGGAAAACCT. *Viperin* Probe: CCAGCG CACAGGGCTCAGGG;

*Pkr* Forward: AGCTGCTGGAAAAGCCACTGA. *Pkr* Reverse: GGGAAACACCATTACTTGTCATAGAC. *Pkr* Probe: AGCTGCTGGAAA AGCCACTGA.

*Socs3* Forward: GCGGGCACCTTTCTTATCC, *Socs3* Reverse: TCCCCGACTGGGTCTTGAC, *Socs3* Probe: CTCGGACCAGCGCCACT TCTTCA.

Reactions were analysed using ABI 7500 Fast Real-time PCR system (Applied Biosystems, USA).

## Flow cytometry

BAL cells were stained with the Live/Dead Fixable Near-IR-Dead Cell staining kit (Invitrogen) for 20 minutes in PBS prior to blockade with anti-CD16/CD32 Fc receptor block (BD Pharmingen) for 20 minutes. Cells were then washed in PBS containing 0.1% sodium azide and 1% BSA followed by staining for surface markers at 4ºC for 30 minutes. Details of antibodies used are shown in Supplementary Table 1. Cells were subsequently washed in PBS containing 0.1% sodium azide and 1% BSA before fixation in 2% paraformaldehyde. Neutrophils were identified using the following surface markers, as previously described[60]: The gating strategy adopted is shown in Supplementary Fig. 5.

## Metabolomics

BAL samples were analysed by Ultrahigh Performance Liquid Chromatography-Tandem Mass Spectrometry (UPLC-MS/MS) performed by Metabolon (Morrisville, NC, USA), as described previously.

Briefly, a Waters ACQUITY UPLC system was coupled to a Thermo Scientific Q-Exactive MS interfaced with a heated electrospray ionization (HESI-II) source and Orbitrap mass analyser operated at 35,000 FWHM mass resolution. Samples were prepared using the automated MicroLab STAR® system (Hamilton, Bonaduz, Switzerland). Several recovery standards were added prior to the first step in the extraction process for QC purposes (see method descriptions below for more detail). To remove protein, dissociate small molecules bound to protein or trapped in the precipitated protein matrix, and to recover chemically diverse metabolites, proteins were precipitated with methanol under vigorous shaking for 2 min (GenoGrinder 2000, Glen Mills, Clifton, NJ, USA) followed by centrifugation. The resulting extract was divided into five fractions: two for analysis by two separate reverse phase (RP)/UPLC-MS/MS methods with positive ion mode electrospray ionisation (ESI), one for analysis by RP/UPLC-MS/MS with negative ion mode ESI, one for analysis by HILIC/UPLC-MS/MS with negative ion mode ESI, and one sample was reserved for backup. Samples were placed briefly on a TurboVap® (Zymark, Clackamas, OR, USA) to remove the organic solvent. The sample extracts were stored overnight under nitrogen before reconstitution in compatible solvents suitable for each analytical protocol. One aliquot was analysed in positive ion mode on a C18 column (Waters UPLC BEH C18-$2.1 \times 100$ mm, $1.7\,\mu m$) using water (Buffer A) and methanol (Buffer B), containing 0.05% perfluoropentanoic acid (PFPA) and 0.1% formic acid (FA). A flow rate of 0.35 mL/min was employed with a linear gradient protocol running from 5% B to 80% over 3.35 min. Seven instrument performance standards (d7-glucose, d5-glutamine, d2-threonine, d5-hippuric acid, d3-methionine, d3-leucine, and Br-phenylalanine) and two process assessment standards (fluorophenylglycine and chlorophenylalanine) were used. A second aliquot was analysed in positive ion mode on the C18 column using 0.05% PFPA and 0.01% FA in water (Buffer A) and 50:50 methanol:acetonitrile (Buffer B). A flow rate of 0.60 mL/min was employed with a linear gradient protocol running from 40% B to 99.5% over 1 min, and held for 2.4 min. Four instrument performance standards (Br-phenylalanine, d5-androstene, d9-progesterone, and d4-dioctyphthalate) and two process assessment standards (d6-cholesterol and chlorophenylalanine) were used.

A third aliquot was analysed in negative ion mode on the C18 column using 6.5 mM ammonium bicarbonate at pH 8 in water (Buffer A) and 95% methanol (Buffer B). A flow rate of 0.35 mL/min was employed with a linear gradient protocol running from 0.5% B to 70% over 4.0 min, rising to 99% B in 0.5 min. Eleven instrument performance standards (d7-glucose, d3-methionine, d3-leucine, d8-phenylalanine, d5-tryptophan, Br-phenylalanine, d15-octanoic acid, d19-decanoic acid, d27-tetradecanoic acid, d35-octadecanoic acid, and d2-eicosanoic acid) and two process assessment standards (tridecanoic acid and chlorophenylalanine) were used. The final aliquot was analysed in negative ion mode using a HILIC column (Waters UPLC BEH Amide $2.1 \times 150$ mm, $1.7\,\mu m$) with a gradient consisting of 10 mM Ammonium Formate, in water:methanol:acetonitrile (15:5:80, Buffer A) and 50% acetonitrile (Buffer B). A flow rate of 0.50 mL/min was employed with a linear gradient protocol running from 5% B to 50% over 3.5 min, rising to 95% B in 2.0 min. Nine instrument performance standards (d35-octadecanoic acid, d5-indole acetate, Br-phenylalanine, d5-tryptophan, d3-serine, d3-aspartic acid, d7-ornithine, and d4-lysine) and two process assessment standards (fluorophenylglycine and chlorophenylalanine) were used. For all sample analysis runs, the MS analysis was performed by within-run switching between MS and data-dependent $MS^n$ scans using dynamic exclusion. The scan range varied slighted between methods but covered $m/z$ 70–1000. Raw data were extracted, peak-identified and QC processed using Metabolon's hardware and software. These systems are built on a web-service platform utilising Microsoft's.NET technologies, which run on high-performance application servers and fibre-channel storage arrays in clusters to provide active failover and load-balancing. Compounds were identified by comparison to library entries of purified standards or recurrent unknown entities. Biochemical identifications were based on three criteria: retention index within a narrow RI window of the proposed identification, accurate mass match to the library ±10 ppm, and the MS/MS forward and reverse scores between the experimental data and authentic standards.

Metabolomics data processing steps were completed using the online Metaboanalyst 5.0 platform (https://www.metaboanalyst.ca/). Data were pre-screened and compounds without identification (i.e. not available within the reference library) or known to be exogenous (e.g. lidocaine) were removed from the dataset. Data were uploaded to the platform as peak area values and features with >50% of missing values were excluded. For compounds where sufficient data points were present, any missing data were imputed as 1/5 of the minimum. A 5% data filter was applied based on IQR values (for non-deviated species) and data were normalised by cube root transformation and pareto scaled. Data normalisation/scaling parameters were optimised and checked before proceeding. A volcano plot comparing up- and down-regulated metabolites in obese vs. non-obese individuals was created using a fold-change threshold of 2.0 and a false discovery rate (FDR) $p$-value of 0.01, assuming an equal group variance. The volcano plot graphic was produced using the data generated by Metaboanalyst and drawn in GraphPad Prism (v10, GraphPad Software, Boston, MA, USA). Unsupervised multivariate analysis was performed by principal components analysis. Supervised multivariate analysis was performed by orthogonal projections to latent structures-discriminant analysis (OPLS-DA). Heat map analyses were conducted by identifying the top 50 correlations (via t-test) and categorised into obese and non-obese individuals. Quantitative Enrichment Analysis (QEA) was performed by inputting the PubChem ID values for all known metabolites alongside a compound concentration table which included the sample phenotype label (i.e. grouping) and metabolite concentration (i.e. peak areas extracted from the metabolomics analyses). Following this, metabolite sets were searched containing at least 2 entries in the SMPDB pathway database. The QEA analysis is performed using the R package globaltest[61] which produces a generalised linear model to estimate a Q-statistic for each metabolite set to describe the correlation between metabolite abundance and clinical grouping. Metabolic enrichments were visualised in order of statistical certainty (i.e. lowest $p$-value first) against a -$\log_{10}$ scale.

## Statistical methods

Data from human obesity studies were analysed using Mann–Whitney $U$ test or Kruskal–Wallis test with Dunn's multiple comparison test. Correlations between datasets were analysed using Spearman's rank correlation coefficient. For animal experiments, group sizes of 5 mice were used and data shown are representative of at least 2 independent experiments. Data were analysed using one- or two-way ANOVA test with Bonferroni's multiple-comparison test. All statistical analyses were performed using GraphPad Prism version 9. Differences were considered significant when $P < 0.05$.

## Reporting summary

Further information on research design is available in the Nature Portfolio Reporting Summary linked to this article.

# Data availability

The data supporting the findings of the study are available in this article and its supplementary Information files, or from the corresponding author on request. Source data are provided with this paper. The metabolomics data generated in this study have been deposited in a database available at https://doi.org/10.17028/rd.lboro.23939772. Source data are provided with this paper.

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

## Acknowledgements

M.A. was supported by a Wellcome Trust/Imperial College Clinical Research Training Fellowship. A.J. is supported by an MRC Clinician Scientist Fellowship (MR/Y000935/1). R.J.S. is a Wellcome Trust Senior Research Fellow in Basic Biomedical Sciences (209458/Z/17/Z). S.L.J. is a National Institute for Health Research (NIHR) Emeritus Senior Investigator and received support from the Asthma UK Clinical Chair (Grant CH11SJ), European Research Council Advanced Grants 233015 and 788575, Medical Research Council Centre Grant G1000758 and Asthma UK Centre Grant AUK-BC-2015-01. A.S. is supported by an MRC Clinician Scientist Fellowship (MR/V000098/1). We thank the staff in the Sir Alexander Fleming Building Flow Cytometry Facility for assistance with flow cytometry experiments. This research was supported by the NIHR Imperial Biomedical Research Centre (BRC). The views expressed are those of the author(s) and not necessarily those of the NIHR or the Department of Health and Social Care.

## Author contributions

M.A., H.A.F., M.R.E, W.S.B, S.L.J. and A.S. conceived, designed, and analysed the human ex vivo studies. M.M.J., R.J.S. and A.S. conceived and designed the animal experiments. J.D. and P.O. conceived and designed the MOSAIC study. K.A.S., K.D.S., A.J.B. and L.M.H. coordinated and analysed metabolomics data. M.A., E.R., P.M., O.M.K and M.R.E performed sampling and experimental work related to human ex vivo studies. M.M.J., O.K., O.P. and A.S. performed experimental work related to the animal studies. A.J. and T.T. performed data analysis related to the MOSAIC study. M.A., H.F., A.J., R.J.S., M.R.E., L.M.H., S.L.J. and A.S. contributed to writing and critical review of the manuscript.

## Competing interests

S.L.J. has personally received consultancy fees from AstraZeneca, Bioforce, Enanta, Myelo Therapeutics GmbH, Bayer, Lallemand Pharma, Synairgen, Novartis, Boehringer Ingelheim, Chiesi, GlaxoSmithKline, and Centocor. S.L.J. is an inventor of patents on the use of inhaled interferons for treatment of exacerbations of airway diseases and on rhinovirus vaccines. S.L.J. is the Director and shareholder of Virtus Respiratory Research Ltd. A.S. has received honoraria for speaking from AstraZeneca. A.J. held a clinical lectureship at the University of Cambridge that was supported jointly by the University of Cambridge Experimental Medicine Training Initiative (EMI) programme in partnership with GlaxoSmithKline (EMI-GSK) and Cambridge University Hospitals NHS Foundation Trust. K.D.S. and K.A.S. are both employees of, and own shares in GSK. PJMO reports grants from the EU Innovative Medicines Initiative (IMI) 2 Joint Undertaking, grants from UK Medical Research Council, GlaxoSmithKline, Wellcome Trust, EU-IMI, UK, National Institute for Health Research, and UK Research and Innovation-Department for Business, Energy and Industrial Strategy; and personal fees from Pfizer, Janssen, and Seqirus. A.J.B. has received consultancy fees from Ammax, Devpro, and Ionis pharmaceuticals, via his institution. The remaining authors declare no competing interests.
