## [Peer Review File · Nature Communications]

Obesity dysregulates the pulmonary antiviral immune responseREVIEWER COMMENTS

Reviewer #1 (Remarks to the Author):

Based on my expertise, I was asked to provide feedback particularly on the metabolomics data and associated analysis.

In this study, the authors describe how obesity changes the immune response in the upper airways when patients are hospitalized with influenza. They cultured different immune cells isolated from patients *ex vivo* and found the macrophages isolated from bronchoalveolar lavage of obese patients to have an impaired immune response. Thus, they performed metabolomics analysis on bronchoalveolar lavage fluid to investigate whether the metabolite profile is altered.

Overall, the metabolomics experiment is reasonable, but it needs some important clarifications.

1. Was the BAL fluid really just dried and reconstituted? It seems like a description of the extraction step is missing.

2. It is unclear which metabolites are shown in Figure 4D. In the results section it says, "hierarchical clustering of the 50 metabolites with the highest concentrations". This raises the question whether a real quantification with external/internal standards was performed, or if this is based on peak areas. The figure legend seems to indicate that the metabolites with the largest difference between the two groups are shown rather than the highest ones. If the selection was based on peak areas/intensities, please rephrase because a higher peak area of a certain metabolite compared to another metabolite does not necessarily mean a higher concentration.

3. The interpretation of Figure 4E (Overview of Enriched Metabolite Sets) is misleading. The way the authors write it, it can be interpreted as if seleno amino acid, methionine, and fatty acid metabolism were upregulated in obese vs. non-obese patients. However, that is not what this analysis is showing. According to MetaboAnalyst, the enrichment ratio is calculated as the number of hits within a particular metabolic pathway divided by the expected number of hits. The authors describe in the method section that the plot shown in 4E was generated with the list of all known metabolites, which indicates that they performed an Over Representation Analysis, meaning it shows how many metabolites of these pathways have been measured, but does not contain any quantitative information. Thus, the interpretation of the plot is wrong. My recommendation would be to remove this plot and any corresponding interpretation as it does not contain any valuable information.

4. The description of the LC-MS methods lacks information:

- The authors mention that the samples were analyzed by Metabolon as previously described. The reference leads to a paper (<https://www.frontiersin.org/articles/10.3389/fcimb.2018.00432/full>) that does not include any details either but refers to another one <https://www.hilarispublisher.com/abstract/categorizing-ion-features-in-liquid-chromatography-mass-spectrometry-metabolomics-data-22363.html> for which I cannot see any full text. Thus, information about exactly what solvents and gradients were used cannot be obtained.

- It is unclear what "a series of standards at fixed concentrations" means. Are these (isotope labeled) internal standards or was a standard addition experiment performed?

- Were MS and MS_n scans collected on all samples? If so, in one run or separate runs?

- Which software was used to process the raw data files?

Thus, in its current form, the metabolomics experiment cannot be reproduced.

5. Currently, no raw data files or source data are available.

6. The authors focused on one of the two metabolites that were more abundant in the obese group. Do the authors have any explanation why so many metabolites were lower in the obese group?

Minor comments

- The vertical lines in the volcano plot in Figure 4C are misleading. According to the methods section and also the colors in the plot a FC cutoff of 2 was used but the lines are at 2 even though $\log_2(\text{FC})$ is shown.

- The resolution of some of the plots is insufficient, for example, it is not possible to read the metabolite names in Figure 4D.

- The heatmap does not show the clustering of the samples, only of metabolites \diamond better refer to it as heatmap instead of hierarchical clustering.

- It is recommended to always start with a PCA to evaluate the data and find potential outliers before performing OPLS-DA. The authors might have done that, but the PCA should be shown as a supplemental figure.
- According to the Methods section, metabolomics analysis was performed on the BAL fluid, but this is not mentioned in the first paragraph of metabolomics results section. Please add to line 194.
- Centrifugation speeds should be given in rcf not rpm.

Reviewer #2 (Remarks to the Author):

In this study, Almond et al explore the impact of obesity on antiviral and proinflammatory responses to influenza virus. This work has clinical implications for both SARS and influenza given that obesity is a well-recognized risk factor for severe disease. Using samples collected from healthy weight or morbidly obese individuals, the authors demonstrate that the airway of obese individuals has an altered metabolomic milieu that impacts macrophage but not epithelial or dendritic cell antiviral responses *ex vivo*. Further, administering exogenous leptin to influenza infected mice significantly reduced inflammation. The authors conclude by demonstrating that obese individuals hospitalized with influenza had increased nasal pro-inflammatory cytokines early in infection. These are compelling data that bring critical new information on the impact of obesity on the pulmonary antiviral response. The experiments are well-controlled and the results support the conclusions.

Comments

1. These studies utilized morbidly obese participants. The authors should comment on the insulin resistance in the cohort as well as use of any drugs known to impact metabolic syndrome, for example metformin, if known.
2. Please comment on any obesity-related differences in cell populations in BAL and how that impacts the findings.
3. Please discuss how your findings may apply to overweight or non-morbidly obese people if known.
4. For the *ex vivo* cell studies, the authors should clarify if these cells were obtained from infected or uninfected individuals.

RESPONSES TO REVIEWER COMMENTS

Reviewer #1 (Remarks to the Author):

Based on my expertise, I was asked to provide feedback particularly on the metabolomics data and associated analysis. In this study, the authors describe how obesity changes the immune response in the upper airways when patients are hospitalized with influenza. They cultured different immune cells isolated from patients ex vivo and found the macrophages isolated from bronchoalveolar lavage of obese patients to have an impaired immune response. Thus, they performed metabolomics analysis on bronchoalveolar lavage fluid to investigate whether the metabolite profile is altered.

Overall, the metabolomics experiment is reasonable, but it needs some important clarifications.

C1: Was the BAL fluid really just dried and reconstituted? It seems like a description of the extraction step is missing.

R1: Thank you for pointing this out and apologies for the original exclusion of the chemical extraction steps within the manuscript. We have updated the section regarding sample preparation to include a more complete description of the process. The updated section now reads as follows:

“Samples were prepared using the automated MicroLab STAR® system (Hamilton, Bonaduz, Switzerland). Several recovery standards were added prior to the first step in the extraction process for QC purposes (see method descriptions below for more detail). To remove protein, dissociate small molecules bound to protein or trapped in the precipitated protein matrix, and to recover chemically diverse metabolites, proteins were precipitated with methanol under vigorous shaking for 2 min (GenoGrinder 2000, Glen Mills, Clifton, NJ, USA) followed by centrifugation. The resulting extract was divided into five fractions: two for analysis by two separate reverse phase (RP)/UPLC-MS/MS methods with positive ion mode electrospray ionization (ESI), one for analysis by RP/UPLC-MS/MS with negative ion mode ESI, one for analysis by HILIC/UPLC-MS/MS with negative ion mode ESI, and one sample was reserved for backup. Samples were placed briefly on a TurboVap® (Zymark, Clackamas, OR, USA) to remove the organic solvent. The sample extracts were stored overnight under nitrogen before reconstitution in compatible solvents suitable for each analytical protocol.”

C2: It is unclear which metabolites are shown in Figure 4D. In the results section it says, “hierarchical clustering of the 50 metabolites with the highest concentrations”. This raises the question whether a real quantification with external/internal standards was performed, or if this is based on peak areas. The figure legend seems to indicate that the metabolites with the largest difference between the two groups are shown rather than the highest ones. If the selection was based on peak areas/intensities, please rephrase because a higher peak area of a certain metabolite compared to another metabolite does not necessarily mean a higher concentration.

R2: Thank you for pointing out this descriptive error in our explanation of the Fig 4D. We have updated multiple sections to better describe what is being shown in this figure section. The updated text reads as follows:

“We therefore profiled BAL fluid metabolite abundances using Ultrahigh Performance Liquid Chromatography-Tandem Mass Spectrometry (UPLC-MS/MS) to gain insight into potential mechanisms that may be driving impaired immunity in obesity (Fig. 4A)... Differential abundance analysis performed via visualisation of the volcano plot identified 17 metabolites with significantly altered responses with the majority (n = 15) downregulated and two metabolites (adenosine

monophosphate (AMP) and glycerol) upregulated in BAL from obese individuals (Fig. 4C). Heat map of the 50 metabolites with the highest peak area responses detected in BAL revealed a similar pattern, with all but three being elevated in non-obese participants, with only AMP, glycerol and mannitol/sorbitol being elevated in obese participants (Fig. 4D).”

C3: The interpretation of Figure 4E (Overview of Enriched Metabolite Sets) is misleading. The way the authors write it, it can be interpreted as if seleno amino acid, methionine, and fatty acid metabolism were upregulated in obese vs. non-obese patients. However, that is not what this analysis is showing. According to MetaboAnalyst, the enrichment ratio is calculated as the number of hits within a particular metabolic pathway divided by the expected number of hits. The authors describe in the method section that the plot shown in 4E was generated with the list of all known metabolites, which indicates that they performed an Over Representation Analysis, meaning it shows how many metabolites of these pathways have been measured, but does not contain any quantitative information. Thus, the interpretation of the plot is wrong. My recommendation would be to remove this plot and any corresponding interpretation as it does not contain any valuable information.

R3: Thank you. The enrichment analysis used for this manuscript was Quantitative Enrichment Analysis (sometimes referred to as QEA) and an Over Representation Analysis was not performed. We understand that the current wording of this suggests that it is assessing the changes in one direction and have updated the descriptions in the results text and figure legend accordingly to reduce the confusion this may have produced.

“Enrichment analysis indicated a range of different metabolites that were enriched within the cohort, including seleno-amino acid, methionine and fatty acid metabolic pathways (Fig. 4E).”

C4. The description of the LC-MS methods lacks information:

- The authors mention that the samples were analyzed by Metabolon as previously described. The reference leads to a paper (<https://www.frontiersin.org/articles/10.3389/fcimb.2018.00432/full>) that does not include any details either but refers to another one <https://www.hilarispublisher.com/abstract/categorizing-ion-features-in-liquid-chromatography-mass-spectrometry-metabolomics-data-22363.html> for which I cannot see any full text. Thus, information about exactly what solvents and gradients were used cannot be obtained.

R4: Thank you for your comment requesting more detail on the analytical protocols. We have updated the section to provide a recent reference where metabolon assessed their assays, and have provided more detailed information regarding the solvents/gradients etc.

“One aliquot was analysed in positive ion mode on a C18 column (Waters UPLC BEH C18-2.1x100 mm, 1.7 µm) using water (Buffer A) and methanol (Buffer B), containing 0.05% perfluoropentanoic acid (PFPA) and 0.1% formic acid (FA). A flow rate of 0.35 mL/min was employed with a linear gradient protocol running from 5% B to 80% over 3.35 min. Seven instrument performance standards (d7-glucose, d5-glutamine, d2-threonine, d5-hippuric acid, d3-methionine, d3-leucine, and Br-phenylalanine) and two process assessment standards (fluorophenylglycine and chlorophenylalanine) were used. A second aliquot was analysed in positive ion mode on the C18 column using 0.05% PFPA and 0.01% FA in water (Buffer A) and 50:50 methanol:acetonitrile (Buffer B). A flow rate of 0.60 mL/min was employed with a linear gradient protocol running from 40% B to 99.5% over 1 min, and held for 2.4 min. Four instrument performance standards (Br-phenylalanine, d5-androstene, d9-progesterone, and d4-dioctylphthalate) and two process assessment standards (d6-cholesterol and chlorophenylalanine) were used. A third aliquot was analysed in negative ion mode on the C18 column

using 6.5mM ammonium bicarbonate at pH 8 in water (Buffer A) and 95% methanol (Buffer B). A flow rate of 0.35 mL/min was employed with a linear gradient protocol running from 0.5% B to 70% over 4.0 min, rising to 99% B in 0.5 min. Eleven instrument performance standards (d7-glucose, d3-methionine, d3-leucine, d8-phenylalanine, d5-tryptophan, Br-phenylalanine, d15-octanoic acid, d19-decanoic acid, d27-tetradecanoic acid, d35-octadecanoic acid, and d2-eicosanoic acid) and two process assessment standards (tridecanoic acid and chlorophenylalanine) were used. The final aliquot was analysed in negative ion mode using a HILIC column (Waters UPLC BEH Amide 2.1x150 mm, 1.7 µm) with a gradient consisting of 10mM Ammonium Formate in water:methanol:acetonitrile (15:5:80, Buffer A) and 50% acetonitrile (Buffer B). A flow rate of 0.50 mL/min was employed with a linear gradient protocol running from 5% B to 50% over 3.5 min, rising to 95% B in 2.0 min. Nine instrument performance standards (d35-octadecanoic acid, d5-indole acetate, Br-phenylalanine, d5-tryptophan, d3-serine, d3-aspartic acid, d7-ornithine, and d4-lysine) and two process assessment standards (fluorophenylglycine and chlorophenylalanine) were used.”

C5: It is unclear what “a series of standards at fixed concentrations” means. Are these (isotope labeled) internal standards or was a standard addition experiment performed?

- Were MS and MSⁿ scans collected on all samples? If so, in one run or separate runs?

- Which software was used to process the raw data files?

R5: Thank you. We have provided more information on which control standards (predominantly labelled isotopes) were used in each experiment. Please see response to C4 above which addresses this request. MS and MSⁿ scans were performed on all runs by scan mode switching. This has been updated in the text to better describe the analytical data collection.

“For all sample analysis runs, the MS analysis was performed by within-run switching between MS and data-dependent MSⁿ scans using dynamic exclusion.”

Raw data files were processed using Metabolon’s in-house software platform. The information regarding this has been expanded within the methods section to provide more detail on this process.

“Raw data were extracted, peak-identified and QC processed using Metabolon’s hardware and software. These systems are built on a web-service platform utilizing Microsoft’s .NET technologies, which run on high-performance application servers and fibre-channel storage arrays in clusters to provide active failover and load-balancing. Compounds were identified by comparison to library entries of purified standards or recurrent unknown entities.”

C6: Currently, no raw data files or source data are available.

R6: Thank you for this comment. We have now linked the source data to a public available repository within the revised manuscript methods section.

C7: The authors focused on one of the two metabolites that were more abundant in the obese group. Do the authors have any explanation why so many metabolites were lower in the obese group?

R7: Thank you for this comment. This finding was unexpected and prior data on the obese versus healthy BAL metabolome is lacking to further validate this. We have added this point to the revised discussion section and further studies are clearly required for independent validation in larger datasets.

“Surprisingly a greater number of metabolites were upregulated in BAL from non-obese subjects. The reasons for this are unclear and further independent validation in larger studies is needed to corroborate our findings.”

C8: The vertical lines in the volcano plot in Figure 4C are misleading. According to the methods section and also the colors in the plot a FC cutoff of 2 was used but the lines are at 2 even though $\log_2(\text{FC})$ is shown.

R8: Thank you for pointing out this descriptive error. Metabolite entries which satisfy the criteria for identification via the volcano plot are chosen based on a minimum FC of 2. However, the resultant output provided by Metaboanalyst incorrectly places the cut-off line at a $\log_2\text{FC}$ of 2.0, where it should be at 1.0 in the x axis and 2.0 in the y axis. We have redrawn the volcano plot using GraphPad Prism in order to correct this coding error. An update to the methods description has been created as follows:

“A volcano plot comparing up- and down-regulated metabolites in obese vs. non-obese individuals was created using a fold-change threshold of 2.0 and a false discovery rate (FDR) p-value of 0.01, assuming an equal group variance. The volcano plot graphic was produced using the data generated by Metaboanalyst and drawn in GraphPad Prism (v10, GraphPad Software, Boston, MA, USA).”

C9 The resolution of some of the plots is insufficient, for example, it is not possible to read the metabolite names in Figure 4D.

R9: Thank you for pointing this out. We have now included an enlarged version of Figure 4D within the supplementary material (new supplementary figure 4) to allow reading of the metabolite annotations. This has been clarified in the figure legend.

C10 The heatmap does not show the clustering of the samples, only of metabolites \diamond better refer to it as heatmap instead of hierarchical clustering.

R10: Thank you for this comment. We have amended the term ‘hierarchical clustering’ to ‘heatmap’, as advised.

C11: It is recommended to always start with a PCA to evaluate the data and find potential outliers before performing OPLS-DA. The authors might have done that, but the PCA should be shown as a supplemental figure.

R11: As recommended, we did begin with an unsupervised PCA analysis and identified a partial separation of groupings which led to the additional investigation using supervised techniques. We have added the PCA scores plot to the supplementary material (Supplementary Figure 2). The methods and results have also been updated as follows:

“Unsupervised multivariate analysis was performed by principal components analysis.”

“Unsupervised Principal Components Analysis demonstrated a partial separation of obese and non-obese groupings (Supplementary Figure 2). This was followed by a supervised analysis where Orthogonal Projections to Latent Structures Discriminant Analysis (OPLS-DA) revealed differences between obese and non-obese individuals (Fig. 4B).”

C12: According to the Methods section, metabolomics analysis was performed on the BAL fluid, but this is not mentioned in the first paragraph of metabolomics results section. Please add to line 194.

Thank you for pointing out our accidental omission of the “fluid” term in this section. We have amended this to read as follows:

“Having observed a specific impairment of antiviral immunity within BAL macrophages from obese individuals, we next reasoned that the metabolomic milieu within the obese airways would be altered and may contribute to the observed ex vivo immune dysregulation. We therefore profiled BAL fluid metabolite abundances using Ultrahigh Performance Liquid Chromatography-Tandem Mass Spectrometry (UPLC-MS/MS) to gain insight into potential mechanisms that may be driving impaired immunity in obesity (Fig. 4A).”

C13: Centrifugation speeds should be given in rcf not rpm.

Thank you, this has been altered.

Reviewer #2 (Remarks to the Author):

In this study, Almond et al explore the impact of obesity on antiviral and proinflammatory responses to influenza virus. This work has clinical implications for both SARS and influenza given that obesity is a well-recognized risk factor for severe disease. Using samples collected from healthy weight or morbidly obese individuals, the authors demonstrate that the airway of obese individuals has an altered metabolomic milieu that impacts macrophage but not epithelial or dendritic cell antiviral responses ex vivo. Further, administering exogenous leptin to influenza infected mice significantly reduced inflammation. The authors conclude by demonstrating that obese individuals hospitalized with influenza had increased nasal pro-inflammatory cytokines early in infection. These are compelling data that bring critical new information on the impact of obesity on the pulmonary antiviral response. The experiments are well-controlled and the results support the conclusions.

C14. These studies utilized morbidly obese participants. The authors should comment on the insulin resistance in the cohort as well as use of any drugs known to impact metabolic syndrome, for example metformin, if known.

R14: Thank you for this important point. There were two individuals in the obese group who had type 2 diabetes mellitus, both of whom were treated with metformin but not insulin. This detail has been added to the revised manuscript (results text and Figure 1).

C15. Please comment on any obesity-related differences in cell populations in BAL and how that impacts the findings.

R15: We thank the reviewer for this important point. Unfortunately we did not carry out flow cytometry of BAL cells to define obesity-related differences in cell populations as part of this study. However, we have retrospectively analysed a transcriptomic dataset of RNA extracted from BAL cells from this study by CIBERSORTx, a computational framework that can infer cell type abundance, using the LM22 signature matrix of 22 blood immune cell types and fifty permutations [reference: Newman et al Nat Biotechnol 2019 PMID 31061481]. Using this method, we observe no

significant differences in the proportion of immune cell populations in BAL between obese and non-obese individuals (see pasted figure below)

C16. Please discuss how your findings may apply to overweight or non-morbidly obese people if known.

R16: We thank the reviewer for this comment. We did not evaluate this group as part of the current study and cannot definitively answer this question. However, given the strong correlation between BMI and serum leptin concentrations reported in multiple studies, we would anticipate that airway leptin concentrations would be similarly correlated with BMI and therefore postulate that there would be a continuum (i.e. the same effect would be present but to a lesser degree) would be observed in terms of antiviral immune suppression. This point has been added to the revised discussion (see below).

'Whether this pathway is similarly dysregulated in overweight or non-morbidly obese individuals remains unknown. Given the strong correlation between BMI and serum leptin concentrations reported in multiple studies, we would anticipate that airway leptin concentrations would be similarly correlated with BMI and therefore we postulate that a continuum (i.e. the same effect would be present but to a lesser degree) would be observed in terms of antiviral immune suppression. Further studies are needed to determine this.'

C17: For the ex vivo cell studies, the authors should clarify if these cells were obtained from infected or uninfected individuals.

R17: Thank you for this comment. Cells were obtained from uninfected individuals for the ex vivo studies and this has been clarified in the revised methods.

REVIEWERS' COMMENTS

Reviewer #1 (Remarks to the Author):

The revised version addresses all my comments.

Reviewer #2 (Remarks to the Author):

Author's were responsive to prior comments.